# Processed Plant-Based Foods for CKD Patients: Good Choice, but Be Aware

**DOI:** 10.3390/ijerph19116653

**Published:** 2022-05-30

**Authors:** Claudia D’Alessandro, Jason Pezzica, Carolina Bolli, Alice Di Nicola, Azzurra Falai, Domenico Giannese, Adamasco Cupisti

**Affiliations:** Department of Clinical and Experimental Medicine, University of Pisa, 56126 Pisa, Italy; jason.pezzica97@gmail.com (J.P.); carolinafrancesca.bolli@aslroma1.it (C.B.); dinicolalice@gmail.com (A.D.N.); azzurra.falai.91@gmail.com (A.F.); domenico.giannese@phd.unipi.it (D.G.); adamasco.cupisti@med.unipi.it (A.C.)

**Keywords:** vegetarian diet, vegan diet, vegan products, processed food, CKD, kidney disease, salt intake, additives, preservatives

## Abstract

The beneficial effects of vegetarian diets are known in the general population and in patients with chronic kidney disease (CKD). In recent years, the market has developed a number of processed plant-based products because of several factors (lifestyle changes, ethical concerns, and sustainability). The composition in terms of nutrients, ingredients, and additives of 560 products available on the market and on online shopping sites was analyzed to understand the characteristics of these products. Processed plant-based meat substitutes have a higher content of salt (+467%), lipids (+26%), mostly unsaturated, and fiber with respect to regular animal-based ones. Protein content is lower (−40%) in plant-based products with respect to corresponding animal ones. Of the 49 additives on the label (on average 2 per product), 20 contain phosphorus, sodium, potassium, or nitrogen. Several plant-based processed products may contain elevated amounts of salt and additives, which make them not optimal for CKD patients. Although a plant-based diet remains a very important tool for CKD nutritional management, patients should be aware regarding the extra content of sodium and additives in processed plant-based products compared to animal-based processed food.

## 1. Introduction

Interest in vegetarianism is increasing in Western countries. Nowadays, many restaurants, company canteens, and school foodservices offer vegetarian meals routinely. The position of the American Dietetic Association and Dietitians of Canada [1] states that vegetarian diets have several nutritional benefits as they provide good quality fibers, carbohydrates, proteins, fatty acids, several minerals such as magnesium, potassium, and antioxidant agents, phytochemicals, vitamin C, and vitamin E [1]. Hence, vegetarian diets, when appropriately planned, may be nutritionally adequate and healthy at any age so play an important role in the prevention and treatment of several chronic diseases [2,3,4].

Currently, data exist regarding the use of plant-based diet also in the treatment of chronic kidney disease (CKD). Dietary patterns characterized by a reduction in animal protein and an increase in plant-origin food such as cereals, legumes, pulses, vegetables, and fruits have several favorable impacts on metabolism and progression of CKD [5].

Moreover, animal protein restriction reduces net acid generation, preventing or correcting metabolic acidosis that elicits protein catabolism, insulin resistance, electrolytes imbalance, and bone damage; metabolic acidosis also represents a factor for the progression of kidney damage [6]. The fiber intake derived from plant foods modulates gut microbiota composition, lowering uremic toxin derived from proteolytic fermentation, and counteracts constipation [7] with a favorable effect on serum potassium control [8]. Evidence also exists that phosphorus from vegetable sources, present in the form of phytate, is less bioavailable in humans allowing a better control of phosphate retention, and its consequences, in CKD [9]. Finally, vegetable proteins have a more favorable impact on glomerular hemodynamics and permselectivity than the same amount of animal protein [10]. 

The plant-food market is continuously growing. Perceived health benefits are the main driver for consumer purchases, while concerns about animal ethics or the environmental impact of animal products are secondary drivers [11,12,13,14,15].

The most used plant proteins are those from soy, peas, chickpeas, spelled, wheat, rice, and corn. There is broad literature on the effect of a vegan/vegetarian diet including position papers of several societies of nutrition of different countries that focus the attention on the positive effects and nutritional adequacy of these dietary patterns. However, they do not take into consideration that a vegetarian diet is not exclusively based on the consumption of natural cereals, legumes, and vegetables. 

A growing aspect is the widespread use of processed plant-based products. Currently, the use of pre-cooked vegan foods has spread widely because they are easy to prepare, even tasty. However, the package remarks “vegan 100%”, “no GMO” (GMO, genetically modified organisms), “organic”, terms that are generally associated to the idea of very healthy natural products.

These products are obtained through food technology processes and, for this reason, need to be supplemented with preservatives, aromas, or dyes because industrial processes deprive them of some characteristics. There are no studies on these products in the scientific literature with the exception of plant-based drinks [16,17,18].

The aim of this survey was to analyze the energy, nutrient, and ingredient composition of a wide range of processed plant-based products and to discuss their use within renal diets.

## 2. Materials and Methods

The present study was conducted through a direct search in supermarkets and discount stores and on the online shopping websites of various large-scale retail food companies. Finally, 560 processed plant-based products were collected. The packaging of each product was carefully analyzed to find the following information:–Presence of claims such as “100% vegan”, “no GMO”, “organic/BIO”, and “gluten-free” or wording generally used to characterize these products as “healthy food”.–Ingredients list to perform a qualitative analysis on the products with particular attention on ingredients that are sources of proteins, and the quality of carbohydrates and fats.–Nutritional facts label to perform a quantitative analysis: energy, fats (saturated or unsaturated), carbohydrates (sugars or starches), fibers, proteins, and salt. The contents of calcium, vitamin B2, vitamin D, vitamin E, iron, iodine, and linoleic acid were also reported for those products in which they were specified, namely vegetable alternatives to milk and cheese.–Presence of additives and the main categories used, namely stabilizers, dyes, acidifiers, thickeners, leavening agents, antioxidants and acidity regulators, emulsifiers, and preservatives. Additives have been specified through the European abbreviation, consisting of the letter “E” followed by three or four numbers as described by the EC 1333 regulation/2008 and subsequently in the EU regulation 1129/2012.

Finally, the average nutritional values obtained from the processed plant-based products analyzed were compared to the corresponding regular ones obtained from the food composition database of the European Institute of Oncology (BDA-IEO) [19]. Data were compared to the Recommended Nutrient and Energy Intake Levels for Italian population for the daily reference serving (LARN IV revision, 2014) [20].

Results are reported as mean ± standard deviation, or median and interquartile range, when applicable. Statistical analysis was performed using a *t*-test for un-paired samples (two-tailed). Differences were considered as significant when *p* < 0.05.

## 3. Results

The processed plant-based product categories investigated in the study are reported in Figure 1.

In total, 65.5% of the examined products had the “100% vegan brand”, 23.9% had the “No GMO” brand, and 45% of products had an “organic” label; 33.21% of the products had the brand or the wording “gluten-free”.

### 3.1. Ingredients 

Figure 2 shows the ingredients present in the plant-derived products. Soy is the most used ingredient, followed by rice, which is frequently used for naturally gluten-free foods production, hence its use is mainly linked to this application. Among starches, in addition to corn and wheat starches, pea, potato, and tapioca starches are also widely used, and modified starch (i.e., additive E1401) serves as a thickener and stabilizer. 

Seed oil is another highly prevalent ingredient, more than four times higher than extra virgin olive oil. 

More than one half of the ultra-processed vegetarian products (especially the meat alternatives) are spiced. The use of spices enhances the taste of the foods, improving their palatability and guaranteeing the consumer a certain degree of satisfaction. Finally, the percentage of sea salt is noteworthy; it is added in the transformation phase of the agri-food chain. This affects the healthiness of these products and the possibility of their use in patients with CKD who must pay attention to the amount and frequency of consumption to control salt intake

### 3.2. Energy and Nutrients

Table 1 summarizes the average energy and nutrient content of the analyzed products per 100 g.

Sauces, dressings, and desserts are rich in calories, while vegetable drinks and yogurts show the lowest energy content.

When the energy content per serving is considered, an increase in vegetable drinks and yogurts (average serving of 200 and 125 g, respectively), and a decrease in sauces and dressings (average servings of 30 and 15 g, respectively) were observed. Energy content in precooked meals ranges from 210 to 500 kcal per serving.

Similarly to energy, fat content shows the same strong discrepancies between groups. A greater variability can be seen in the categories of sauces, dressings, and desserts. Fat content is almost the same for meat substitutes, precooked meals, and cheeses (as average 8.4–11.9 g for burgers, cod cuts, cutlets, meatballs; 16 g for the cheese group), and it is quite high; 100 g of adult beef fillet provides 5 g fat/100 g, corresponding to half of the same serving of the processed plant-based alternatives. The amount of saturated fatty acids is always very low, except for the cheese group, where it increases considerably up to an average of 13.6 g of saturated fatty acids per 100 g. In any case, the average lipid content is high, in particular for cheeses which provide an average of 8.5 g of saturated fatty acids per serving.

Regarding sugar content, higher amounts are recorded for the yogurt group (10 g/100 g of product, 12.5 g/serving) and for that of desserts and ice creams (as average 24 g/100 g of product with a maximum of 77 g/100 g of product). The highest median carbohydrates content per serving is observed for precooked meals (26.4 g per serving, of which 3.5 g sugars), even with respect to that of desserts (20 g/portion, of which 13 g sugars); cold cuts, vegetable drinks, sauces, and dressings have the lowest content of carbohydrates (from 0.9 to 2.9 g per serving). 

The median protein content per 100 g of products is 25.5 g for processed plant-based meat, 14 g for meatballs, 12.8 g for burgers (12.8 g/100 g), and 11.7 g for cutlets and nuggets. The protein content in precooked meals is extremely variable according to the ingredient used for their preparation, reaching up to 52 g of protein/100 g. Additionally, cheeses have a variable protein content with the highest value in tofu, 16 g/100 g of products. In the remaining food categories, protein content varies from 0.3 to 3.1 g/100 g of product.

Protein content per serving varies as follows: burgers 7.2–16 g; plant-based meat substitutes 10.9–16.3 g; cutlets and nuggets 6.6–13.0 g, meatballs 6.7–14.9 g. Overall, the average protein content of plant-based alternatives to meat are lower than those derived from regular meat and poultry products. In fact, red meat (fillet of adult bovine) provides about 20.5 g of protein/serving and poultry (chicken breast, skinless) about 23.3 g/serving. 

As expected, plant-based products have a high content of fibers. Plant-based alternatives to meat provide an average of 3.5–5 g of fiber per 100 g, which is absent in regular meat. The groups with lower amount are cheeses, desserts, and dressings with average values up to a maximum of 3.5 g/100 g, but up to maximum values of 16.3 g of fiber per 100 g of coconut butter whose serving, however, is 10 g. The average fiber content per serving is 4.2 g for burgers, 3.8 g for the group of cutlets and nuggets, and 4.2 g for plant-based meatballs up to a median of 5.5 g for precooked meals, up to 32 g per serving

Eight out of the 11 food categories investigated in this study have a high content of salt. The groups with the lowest salt content per 100 g of product are vegetable drinks, yogurt, and desserts. All plant-based alternatives to meat and cheese have a high salt content, with medians of 1.2–1.3 g up to 1.8 g/100 g in precooked meals, where salt is used as a flavor enhancer and as a preservative, providing a median amount of 1.1 g/100 g. Sauces and dressings are as rich in salt as any similar product available in any supermarket. Salt content is still high also when considered per serving. Precooked meals remain a significant source of salt, 1.3–3.4 g per serving up to 5 g found in one of the studied products. Vegetable alternatives to the meat and cheese groups have 1.0 and 1.2 g (corresponding to 389–472 mg of sodium) per serving. Salt content in vegetable drinks and yoghurts is similar to that of regular animal-based foods. For example, partially skimmed UHT cow’s milk provides about 84 mg of sodium per serving, compared to a median of 79 mg of sodium for the vegetable alternative. White yogurt derived from partially skimmed milk, according to the IEO database, provides on average 56 mg of sodium, compared to 74 mg provided by the vegetable alternative.

### 3.3. Plant-Based Processed Products vs. Animal-Based Foods 

Figure 3 and Figure 4 show a comparison between plant-based processed products and animal-based analogues. What emerges is, on average, a lower protein, higher fat and salt content in plant-based products than corresponding regular ones.

What makes a positive difference between plant-based and animal-based products is the higher fiber content in the former that ranges from 3.5 to 4.5 g per serving. 

Plant-based cold cuts and sausages show little difference in comparison to animal-based ones even if the former should be considered a better choice as they bring less saturated fatty acids and salt and greater amounts of fiber than the latter.

Vegetable drinks, cheeses, yogurts, dessert, and ice cream show very little difference with the corresponding animal-based products. For example, all cheeses (both vegetable and regular ones) have overlapping amounts of salt and saturated fatty acids.

### 3.4. Plant-Based Processed Products and Additives

Each processed plant-based product contains 1.84 additives on average. It follows that what is advertised as “healthy” and “organic” is often subjected to chemical manipulations, carried out in order to ensure a proper conservation and a good appearance. 

The bar graph in Figure 5 represents the 11 food categories analyzed, placed in descending order according to the variable percentage of additives per category.

It was found that 43.2% of total additives are used to produce desserts, ice cream, and yogurt, while the other 56.8% is distributed into products belonging to the other nine food categories. Figure 6 shows the “more added” products: desserts and ice creams are the most processed with an estimated average of 3.0 additives per product, followed by yogurts with an average of 3.0 additives per product, and vegetable drinks, 2.3 additives per product. The analysis of food labels shows that the less processed products are plant-based meat substitutes (especially burgers and meatballs) with estimated averages of 0.8 and 0.7 additives per product, respectively.

As a whole, 49 additives were detected, with a different prevalence in each category (Figure 6) for a total of 683 times among all additives. The most used (>50%) are those belonging to the class of stabilizers/thickeners, followed by emulsifiers and, finally, by antioxidants. All the other classes were used in smaller percentages (<7%). 

## 4. Discussion

The present study on processed plant-based products showed a high content of simple sugars and fats together with a high sodium content, and an almost constant presence of additives.

Ingredients such as sugar, oil, and flavorings are added to the plant-based substitute to make them more palatable and more acceptable to consumers [16].

Processed plant-based products have spread widely because they are tasty and easy to prepare, because of cultural/ethical/religious reasons, and because they are perceived as healthier and more sustainable [21].

In total, 65.5% of the examined products reported the “100% vegan” brand being suitable for vegan subjects; 23.9% had the “No GMO” brand, probably the use of ingredients free from genetic modifications consisting of a method of propaganda of a food model often associated with a healthy dietary pattern. However, processed foods bring with them many concerns derived from methodologies applied to improve their conservation and palatability. In total, 45% of the studied products have an “organic” label: these products are declared as derived from organic farming and this, like the No GMO products, is a further method to advertise the healthiness of these plant-based products. A total of 33.2% of the products have the brand or the wording “gluten-free”, so they are suitable for subjects suffering from celiac disease: these foods, in fact, can be freely consumed by celiac subjects as they are produced with naturally gluten-free cereals (rice, millet, sorghum, buckwheat, etc.) or with soy without the addition of wheat starch.

Regarding the ingredients used in processed plant-based products, the study shows that soy is the most used, and it is often declared on the label as the first or second ingredient. Then comes rice, which is frequently used for naturally gluten-free food production. Among starches, in addition to corn and wheat starches, pea, potato, and tapioca starches are also widely used, as well as modified starch (additive E1401), used as a thickener and stabilizer. Seed oil is another highly represented ingredient. The large number of products containing oil (even in vegetable drinks and yogurts) explains the high energy content of each food category examined. Oil is used not only to homogenize and make the product greasy, but also as a preservative, especially in precooked foods. Over half of the ultra-processed vegetarian products (especially meat alternatives) are spiced. The use of spices enhances the taste of these foods, improving their palatability and guaranteeing the consumer a certain degree of satisfaction. Finally, the percentage of sea salt is noteworthy; it is added in the transformation phase of the agri-food chain. This affects the healthiness of the products and the possibility of their unrestricted use in patients with CKD who must pay attention to the amount and frequency of consumption to control salt intake.

The average carbohydrate content per 100 g of product is high for all categories, but this is expected as the basic ingredients for these processed products are cereals and legumes. This also explains the high fiber content in respect to the corresponding animal-based regular foods. It is noteworthy that in some product fibers are also added for technical reasons (for example, citrus, fruit, pea fiber, or psyllium) so that the total content is given by the fibers naturally contained plus fibers added during the production procedure. Fat content in plant-based burgers, meatballs, and cutlets is more than double with a prevalence, however, of unsaturated fatty acids. This result is expected given that plant-based products are produced with plant sources. The difference in fat content among categories is reduced considering fat content per serving. In plant-based products, proteins were derived mainly from legumes (soy, chickpeas, peas, and lentils): the amount is high but roughly halved compared to regular meat products.

Precooked meals provide the highest fat content per serving while vegetable drinks and yogurts show the lowest. The analysis revealed that plant-based meat substitutes have a very high salt content, up to almost seven times compared to the regular meat products. The lack of data from the literature prevents from making comparisons. The only data available at present concerns plant-based milks where a high content of salt and simple sugars has been highlighted [16,18].

Plant-based precooked meals provide a variable energy content ranging from 210 to 500 kcal per serving. This is like the energy content of animal-based precooked analogues. Therefore, the general population should be warned because sometimes they buy these products because they consider them low in calories. Regarding the CKD population, a further warning must be added because, like most precooked meals, they are rich in salt. Moreover, the presence of phosphorus, sodium, or potassium preservatives represents a further aspect that requires attention.

This widespread presence of additives in the analyzed products was unexpected because they carry various claims that tend to highlight the characteristics of organic, vegan, non-GMO products that lead to the idea of “healthy food”. It is noteworthy that out of the 49 additives shown on the label (on average 2 per product), 20 contain phosphorus, sodium, potassium, and nitrogen in unspecified amounts. Unfortunately, this information cannot be found under the current regulations on the use of additives and labeling. In fact, producers are not required to add these data probably because additives are used also to guarantee safety and preserve the properties and appearance of a product destined to remain on the market shelves for a long time. The most used are stabilizers/thickeners and antioxidants because they have the task of extending storage times, and emulsifiers, which are used to ensure greater homogeneity, allowing the emulsion of naturally immiscible parts.

## 5. Conclusions

In conclusion, processed plant-based products represent an interesting chance to simplify and vary the diet of CKD patients who benefit from vegetarian diets [22,23]. However, attention must be paid to servings and frequency of consumption of these products in order not to exceed with salt and additives intake. This may not be a concern in the general population, but it could instead represent a limitation for free, unrestricted use of these products in patients with CKD.

## Figures and Tables

**Figure 1 ijerph-19-06653-f001:**
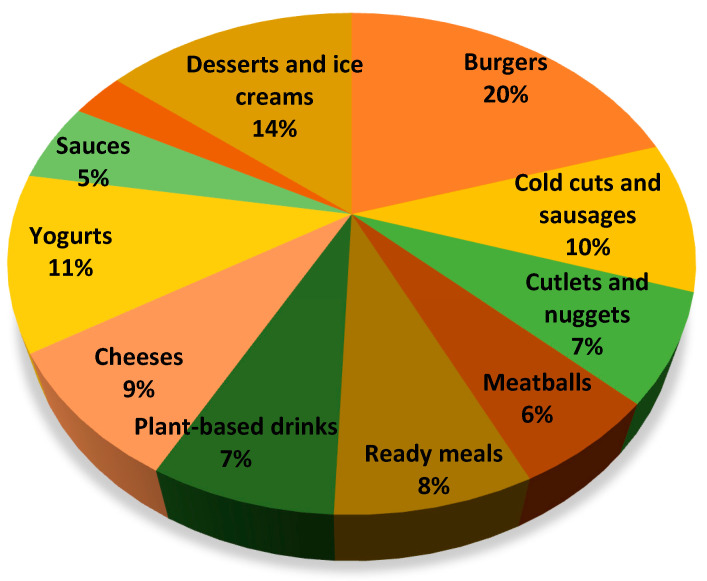
Prevalence of processed plant-based products categories examined in this study. The graph reported the percentage of samples in each categories considering a total number of 560 analyzed products.

**Figure 2 ijerph-19-06653-f002:**
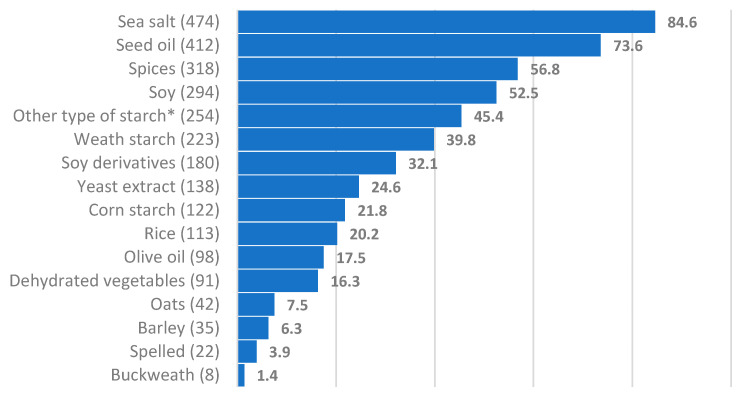
Ingredient prevalence expressed as percentage of products in which an ingredient appears. * Potato, tapioca, pea, starches, and modified starches (additive E1401), used as a thickener and as a stabilizer. Data represent the frequency with which an ingredient appears in the analyzed products. The total number of analyzed products per category is reported in brackets.

**Figure 3 ijerph-19-06653-f003:**
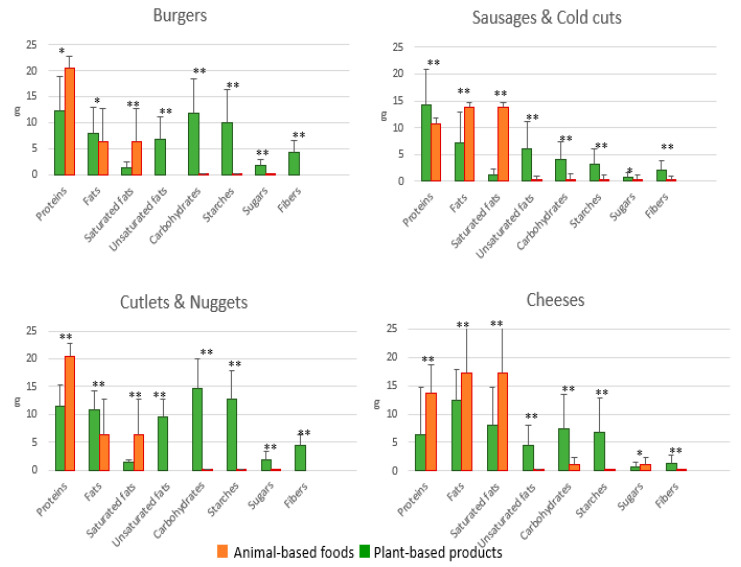
Macronutrients and fiber composition of processed plant-based products vs. regular animal-based foods * *p* < 0.05; ** *p* < 0.01. All the data were presented as mean ± SD.

**Figure 4 ijerph-19-06653-f004:**
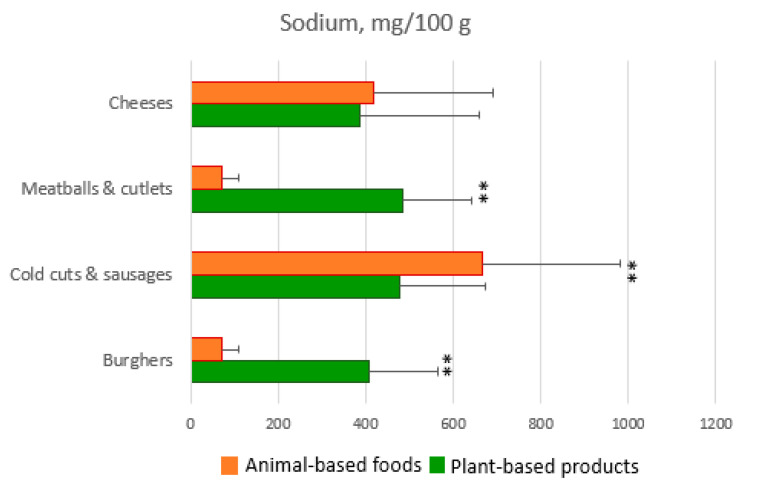
Sodium content of processed plant-based products vs. regular animal-based products. ** *p* < 0.01. All the data were presented as mean ± SD.

**Figure 5 ijerph-19-06653-f005:**
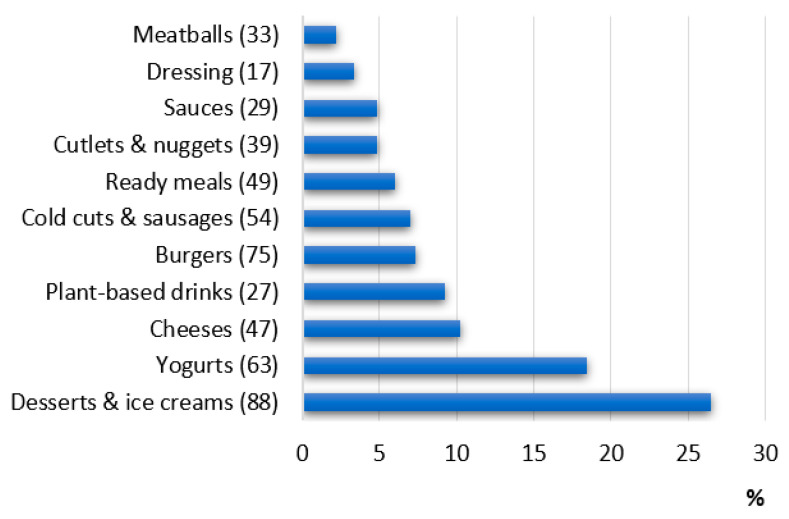
Prevalence of additive-containing products in the categories of processed plant-based products. The total number of analyzed products per category is reported in brackets.

**Figure 6 ijerph-19-06653-f006:**
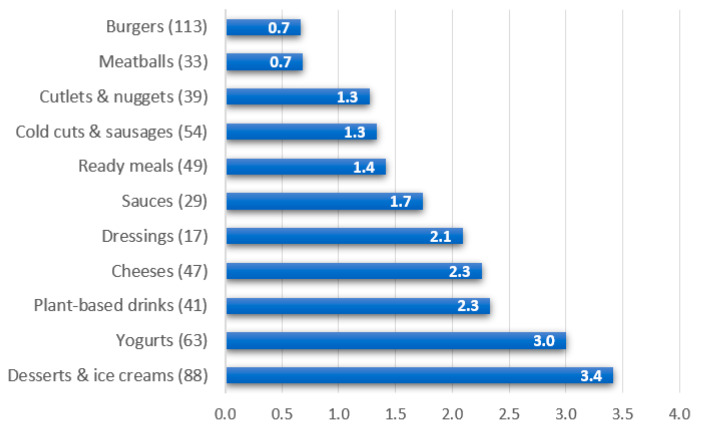
Average number of additives per product in each category of the processed plant-based products included in the study. The total number of analyzed products per category is reported in brackets.

**Table 1 ijerph-19-06653-t001:** Energy and nutrient content per 100 g of processed plant-based products analyzed in this study. All the data are presented as mean ± SD. The total number of analyzed products per category is reported in brackets. ^§^ Cold cuts include also sausages; ° Cutlet include also nuggets; °° Dessert include also ice creams; FA: fatty acids.

Plant-Based Products(n)	Energy kcal	Proteins g	Fats g	Saturated FA g	Unsaturated FA g	Carbohydrates g	Starches g	Sugars g	Fibers g	Salt g
Burger (113)	178 ± 50	12.3 ± 6.6	8.0 ± 4.9	1.3 ± 1.3	6.7 ± 4.3	11.9 ± 6.4	10.1 ± 6.2	1.8 ± 1.2	4.3 ± 2.3	1.0 ± 0.4
Cold cuts ^§^ (54)	142 ± 63	14.8 ± 6.8	7.3 ± 5.6	1.1 ± 1.2	6.2 ± 4.9	4.1 ± 3.3	3.2 ± 2.8	0.9 ± 0.7	2.1 ± 1.9	1.2 ± 0.5
Cutlet ° (39)	196 ± 42	10.1 ± 3.9	9.4 ± 3.3	1.2 ± 0.6	8.2 ± 3.1	15.7 ± 5.4	13.8 ± 5.0	1.9 ± 1.5	3.5 ± 1.8	1.2 ± 0.4
Meatballs (33)	211 ± 44	11.4 ± 4.3	10.9 ± 3.5	1.4 ± 0.4	9.5 ± 3.3	14.6 ± 8.3	12.8 ± 8.1	1.8 ± 1.1	4.5 ± 2.0	1.2 ± 0.4
Cheese (47)	170 ± 58	6.4 ± 8.3	12.4 ± 5.5	7.9 ± 6.7	4.5 ± 3.6	7.4 ± 6.1	6.8 ± 6.0	0.6 ± 0.9	1.4 ± 1.3	0.98 ± 0.7
Yogurt (63)	94 ± 14	4.2 ± 1.0	2.7 ± 0.8	0.2 ± 0.9	2.5 ± 2.8	12.6 ± 4.7	1.0 ± 0.7	11.6 ± 4.3	0.9 ± 0.6	0.2 ± 0.1
Dessert °° (88)	168 ± 65	2.5 ± 1.4	7.6 ± 4.1	4.0 ± 3.3	3.6 ± 2.2	21.6 ± 10.2	7.6 ± 7.4	14.0 ± 7.6	1.3 ± 1.3	0.1 ± 0.1
Drinks (41)	56.2 ± 18	1.9 ± 1.6	2.2 ± 1.1	0.3 ± 0.3	1.9 ± 1.0	7.0 ± 5.2	2.2 ± 2.8	4.9 ± 3.5	0.5 ± 0.5	0.1 ± 0
Ready meals (49)	320 ± 172	19.0 ± 18	18.0 ± 13.7	2.5 ± 1.8	15.5 ± 12.3	17.0 ± 12.0	13.4 ± 10.5	3.7 ± 3.5	6.5 ± 5.5	1.8 ± 1.1
Sauces (29)	106 ± 49.0	2.6 ± 1.6	7.1 ± 6.5	1.3 ± 1.8	5.9 ± 5.6	7.6 ± 6.3	1.1 ± 0.8	6.5 ± 6.2	0.5 ± 0.5	0.4 ± 0.2
Dressings (17)	94.0 ± 57.1	1.2 ± 1.1	8.7 ± 5.8	1.8 ± 2.1	6.9 ± 5.3	2.9 ± 2.6	1.1 ± 1.2	1.9 ± 2.4	0.3 ± 0.4	0.4 ± 0.3

## Data Availability

All data generated or analyzed during this study are included in this published article.

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
