# Peer review of "Processed Plant-Based Foods for CKD Patients: Good Choice, but Be Aware"

_ijerph, 2022, doi:10.3390/ijerph19116653_

Round 1

Reviewer 1 Report

Thank you for the invitation to review the manuscript.

 The manuscript of  D’Alessandro   et al. describes the comparative study of  processed plant-based meat substitutes in vegetarian diets of  patients with chronic kidney disease (CKD).

 I have found the manuscript interesting and comprehensive. The authors are specific to the relevant objectives of the study. All the assays are correctly described and informed. The paper can be an important tool for CKD nutritional management.

I have a few suggestions for authors:

  • Statistical methods have described the test and significance levels, please add on each figure "All the data were presented as Mean/SD or SE".
  • The discussion section is well managed, however, the authors could describe other representative studies in order to underscore its significance.
  • The results should be compared with other data from the literature, if any.
  • I did not find the bibliography

Author Response

The manuscript of  D’Alessandro   et al. describes the comparative study of  processed plant-based meat substitutes in vegetarian diets of  patients with chronic kidney disease (CKD).

 I have found the manuscript interesting and comprehensive. The authors are specific to the relevant objectives of the study. All the assays are correctly described and informed. The paper can be an important tool for CKD nutritional management.

 I have a few suggestions for authors:

  • Statistical methods have described the test and significance levels, please add on each figure "All the data were presented as Mean/SD or SE".
  1. Thanks for your comments, we have added the suggested sentence in the figures

  • The discussion section is well managed, however, the authors could describe other representative studies in order to underscore its significance.
  1. This investigation started just after realizing that there is a lack of information about these processed products. Scientific literature exists reporting the benefit of vegan diets and natural plant based food but, to our knowledge, no data have been found regarding the processed plant-based products. We only found data on plant based drinks confirming that salt is the most commonly added ingredient.

More references have been added regarding plant-based drinks in the references section

  • The results should be compared with other data from the literature, if any.
  1. As we commented above currently there are few data about the processed products except from plant based drinks. We have added some references regarding plant based beverages in the reference list and a comment in the discussion section.

  • I did not find the bibliography
  1. The reference section now lists 23 articles and it follows the discussion section

Reviewer 2 Report

The manuscript entitled “Processed Plant-Based Foods for CKD Patients: Good, but Not with Closed Eyes” provides important and insightful perspectives on plant-based diets and their ingredients.

Manuscript should be revised based on following comments:

1) Figure 1. Should also provide percentage which will make it easier to understand. The total number of samples should also be provided.

2) product categories in Figure 1 and Table 1 are different. They should be same so that analysis will be better.

3) There are many typographical errors like Vegetal in stead of vegetable, Burgher (table 1) instead of burger, etc. which should be corrected.

4) Figure 3, does cold cuts also cover sausages as in Figure 1?

5) The resolution of Figure 3 should be increased.

6) Figure legends use * or **  for significance but they are not represented inside figure.

7) Values provided in Table 1 and Figures 2 to 6 seem to be values calculated as average of certain number of products. The sample size (n) should be provided for each of them along with +-SD.

Author Response

Reviewer 2

The manuscript entitled “Processed Plant-Based Foods for CKD Patients: Good, but Not with Closed Eyes” provides important and insightful perspectives on plant-based diets and their ingredients.

Manuscript should be revised based on following comments:

1) Figure 1. Should also provide percentage which will make it easier to understand. The total number of samples should also be provided.

  1. Percentage and total numbers of products have been added in figure 1

2) product categories in Figure 1 and Table 1 are different. They should be same so that analysis will be better.

  1. Thank you. Accordingly, Table 1 was modified in the text and now it reports the same categories as in Figure 1

3) There are many typographical errors like Vegetal in stead of vegetable, Burgher (table 1) instead of burger, etc. which should be corrected.

  1. We apologizes for the inaccuracy, we checked the text for typing errors

4) Figure 3, does cold cuts also cover sausages as in Figure 1?

  1. Yes, it is: cold cuts are together with sausages.

5) The resolution of Figure 3 should be increased.

  1. We tried to improve Figure 3 resolution. I hope now is good enough.

6) Figure legends use * or **  for significance but they are not represented inside figure.

  1. We apologizes for this inadvertence, we have included the symbols for significance in the figures

7) Values provided in Table 1 and Figures 2 to 6 seem to be values calculated as average of certain number of products. The sample size (n) should be provided for each of them along with +-SD.

  1. Thank you for your note. Table 1, Figures 2 to 6 have been reviewed. As regard the Table 1 we added all the analyzed categories, reporting mean ±SD, and the total number of products per category. Figure 2 represents the frequency with which an ingredient appears in the analyzed products. The total number of analyzed products per each category is now reported in brackets. Figure 6 shows the average number of additives per category o studied product. We added (in brackets) the total number of products analyzed

Reviewer 3 Report

GMO, must be defined.

In my opinion, the authors should rewrite the objective. The authors did not evaluate the beneficial impact of these products, only inferred possible disadvantages

I consider the article well written and of scientific importance. The problem is related to the objective: "The aim of this survey was to analyze the energy and nutrients composition of a wide 62 range of processed plant-based products and to evaluate their more or less beneficial impact in the nutritional treatment of patients with CKD." I consider this should be rephrased. The authors did not evaluate the impact of these products in patients with CKD thay only inferred the potential harms.

Author Response

Comments and Suggestions for Authors

-GMO, must be defined.

  1. The acronym GMO ” (genetically modified organisms) has been now defined and inserted in the text

-In my opinion, the authors should rewrite the objective. The authors did not evaluate the beneficial impact of these products, only inferred possible disadvantages

  1. We appreciated your concerns but in this paper we have not at all questioned the benefits of vegetarian diets or of natural plant products. Only the processed vegetable products are at issue, namely those products obtained through food technology processes and those which need to be added with preservatives, aromas or dyes because industrial processes deprive them of some characteristics. There are no studies on these products in the scientific literature with the exception of plant-based drinks (we added some references on plant-derived drinks in the references section). The take-home-message we have reported in the conclusions is not that they are not good for CKD patients, but simply that patients with CKD must be a little bit careful.

“In summary, processed plant-based products represent an interesting chance to make easier and varied the diet in CKD patients who benefit from vegetarian diets [22, 23]. However, attention must be paid to servings and frequency of weekly consumption of these products in order not to exceed with salt and additives intake. This may not be a concern in the general population, but it could instead represent a limitation for a free, unrestricted use of these products in patients with CKD”

We have also modified the title, in order to endorse our conclusion which is not meant to be negative but only to encourage more attention on Processed Plant-Based Foods for CKD Patients

Manuscript title has been changed as follow “Processed Plant-Based Foods for CKD Patients: good choice, but be aware”

New references:

Silva ARA, Silva MMN, Ribeiro BD. Health issues and technological aspects of plant-based alternative milk. Food Res Int. 2020 May;131:108972. doi: 10.1016/j.foodres.2019.108972.

Mäkinen OE, Wanhalinna V, Zannini E, Arendt EK. Foods for Special Dietary Needs: Non-dairy Plant-based Milk Substitutes and Fermented Dairy-type Products.Crit Rev Food Sci Nutr. 2016;56(3):339-49

Fructuoso I, Romão B, Han H, Raposo A, Ariza-Montes A, Araya-Castillo L, Zandonadi RP. An Overview on Nutritional Aspects of Plant-Based Beverages Used as Substitutes for Cow's Milk. Nutrients. 2021

-I consider the article well written and of scientific importance. The problem is related to the objective: "The aim of this survey was to analyze the energy and nutrients composition of a wide range of processed plant-based products and to evaluate their more or less beneficial impact in the nutritional treatment of patients with CKD." I consider this should be rephrased. The authors did not evaluate the impact of these products in patients with CKD thay only inferred the potential harms.

  1. Thanks for your comments and suggestions. We agree with your criticism: the aim has been rewritten as follows:

“The aim of this survey was to analyze the energy, nutrients and ingredients composition of a wide range of processed plant-based products and to discuss their use within the renal diets”

Round 2

Reviewer 2 Report

Authors have revised the manuscript as suggested.